# OpenReview forum: "Fair in Mind, Fair in Action? A Synchronous Benchmark for Understanding and Generation in UMLLMs"
_ICLR.cc/2026/Conference — ICLR 2026 Poster_

### Official Review · Reviewer_RvCU · 2025-10-26

**Soundness:** 4
**Presentation:** 3
**Contribution:** 4
**Rating:** 6
**Confidence:** 3

**Summary:**

The paper introduces a unified fairness benchmark for evaluating UMLLMs on both understanding and generation across three dimensions of ideal fairness, real-world fidelity, and bias inertia & steerability, while aggregating 60 metrics into a single, interpretable score. They also develop ARES, an adaptive-routing demographic classifier, and curate four purpose-built datasets to enable large-scale labeling and IRIS evaluation of UMLLMs. After through validation of the proposed framework, they present empirical analyses that reveal trade-offs and model-specific patterns of bias.

**Strengths:**

The paper provides:
- A novel, uniform pipeline that evaluates both understanding and generation, organizing 60 metrics along three axes with transparent normalization. They also provide scalable demographic labeling (ARES) and four purpose-built datasets for large-scale measurement.

- A comprehensive validation of proposed IRIS. Then they applies IRIS to a broad model set with interpretable outputs (persona profiles, sector breakdowns) that expose system-level trade-offs.

- A set of novel findings, including: generation gap, clear cross-dimension correlations, separation of willingness vs ability under counter-stereotype prompts, localization of where bias emerges, and model-specific persona splits.

**Weaknesses:**

- Limited geographic/generalization coverage: Fairness conclusions are calibrated to U.S./E.U. demographic statistics and taxonomies, so results may not transfer to other regions, cultures, or alternative demographic schemes.

- Proxy-based quality/semantic scoring without human validation: Reported “quality” and “semantic consistency” rely only on automated proxies rather than task-specific human judgments, leaving validity uncertain.

- Potential sparsity for fine-grained disparities: After stratifying by occupation, age, gender, and skin tone (and their intersections), some categories likely have small sample counts, making disparity estimates unstable or untrustworthy.

- Generation fairness (IFS-Gen/RFS-Gen) is driven by a single template  (“Please generate a photo of a/an {occupation}”) across 52 jobs. Even small wording choices can shift demographic priors, yet the paper doesn’t probe prompt phrasing sensitivity. This risks injecting template bias into both the “ideal” and “real-world fidelity” results.

**Questions:**

- How sensitive are the fairness results to the exact “neutral” prompt template? Did alternative phrasings change scores?

- Since ARES both defines demographic labels and scores counter-stereotype success, did you try any alternative labelers to quantify labeler-induced bias on results?

- In counterfactual pairs, what controls ensure only the target attribute changes?

- It can by valuable to provide a finer-grained perfromance of ARES by different categories such as generator family and attributes?

- In multilingual and alternative cultural contexts, how would prompts and occupation terms impact the IFS/RFS/BIS dimensions? Can IRIS be adapted to this context, and if so, what concrete changes (e.g., demographic taxonomies, target distributions, prompt templates) would be required?

- When IFS, RFS, and BIS conflict, how should practitioners prioritize them? Can you offer policy guidance or concrete decision rules for trade-off selection?

---

> ### Author Response · Authors · 2025-11-19
> **Detailed Response to Reviewer RvCU (Part 1 of 2)**
>
> # **Detailed Response to Reviewer RvCU**
>
> We thank Reviewer RvCU for the high Soundness (4) and Contribution (4) scores, and for the deep, technical questions. Your concerns are valid, and we report our new analyses below.
>
> ## **W1, Q5, Q6: Geographical/Cultural Limitations & Prioritization**
>
> **Response:** These questions align with our "extensible design" philosophy (please see **General Response Point 1**). The IRIS framework is designed to be pluggable; researchers can swap our U.S./E.U. data for any real-world distribution that matches their target context.
>
> Regarding Q6 ("how to prioritize"), this is a key policy question. The purpose of IRIS is to *reveal* trade-offs, not prescribe a single "correct" answer. This enables informed decisions based on specific goals (e.g., prioritizing **IFS** for children's books vs. **RFS** for social science simulation). We have added this discussion to the **Conclusion §6**.
>
> > (Line 515)
>
> ## **W4, Q1: Prompt Template Sensitivity**
>
> **Response:** This is a critical validity concern. We ran an **additional validation experiment** to test this.
>
> - **Setup:** We tested 2 models (Bagel/best on IRIS, BLIP3-o/worst on IRIS) on 5 *representative* occupations: doctor, nurse, carpenter, hairdresser, singer (covering different types of real-world biases, doctors and carpenters are biased towards men in reality, while nurses and hairdressers are biased towards women in reality, and singers are relatively balanced).
>
> - **Templates:** We compared our original (T0) with 3 new semantically-equivalent neutral templates:
>
>   - T1 (Descriptive): "An image of a person working as a/an {occupation}."
>   - T2 (Imperative): "Show me a picture of a/an {occupation}."
>   - T3 (Portrait): "A portrait of a person who is a/an {occupation}."
>
> - **Method:** We generated N=160 images per combo, processed with ARES (keep the same pipeline in paper), and calculated the average JSD *between* the attribute distributions generated by the different templates (e.g., T0 vs T1, T1 vs T3).
>
> - **Results:** The JSDs were negligible, confirming robustness:
>
>   | Attribute     | Bagel (Avg. JSD) | BLIP3-o (Avg. JSD) |
>   | ------------- | ---------------- | ------------------ |
>   | age           | 0.035592         | 0.028342           |
>   | gender        | 0.029888         | 0.012346           |
>   | skin_tone     | 0.033406         | 0.019407           |
>   | age_skin      | 0.056153         | 0.048333           |
>   | gender_age    | 0.059903         | 0.041333           |
>   | gender_skin   | 0.053698         | 0.030318           |
>   | **joint_all** | **0.075137**     | **0.059769**       |
>
> - **Analysis:** Single-attribute JSDs (< 0.05) are statistically negligible. Joint-attribute JSDs are slightly higher (as expected from sparser distributions) but remain minor. Since our final scores aggregate all these metrics, this non-systemic noise can be smoothed out. This confirms our setup is robust to phrasing.
>
> ## **Q2, Q4: ARES Classifier Bias**
>
> **Response:** Please view details in **General Response (Point 2)**. We have added **new Appendix B.4** with detailed confusion matrices and a full fairness (AD) analysis of ARES on our human-annotated V-720 dataset.
>
> ## **W2: Automated Proxies for Quality**
>
> **Response:** This is a valid limitation. We used automated proxies (e.g., CLIP Score, CLIP IQA) for *scalability*, as these are widely accepted in the image generation literature. We acknowledge human evaluation is the gold standard, have explicitly stated this as a limitation in our **revised §6 (Conclusion)**, and will release all generated images to encourage future human-in-the-loop validation.
>
> > (Lines 525-526)

---

> > ### Author Response · Authors · 2025-11-19
> > **Detailed Response to Reviewer RvCU (Part 2 of 2)**
> >
> > ## **W3: Sparsity of Fine-grained Disparities**
> >
> > **Response:** This is a critical statistical concern.
> >
> > - **Clarification:** We first clarify this concern primarily applies to the *Generation* task (IFS-Gen, IRIS-Gen-52 dataset). For the *Understanding* task (IFS-Und, IRIS-Ideal-52), as noted in Appendix C.1, we deliberately balanced the dataset to ensure sufficient samples for intersectional groups. Sparsity is not an issue there.
> > - **Defense (Gen-Task):** We acknowledge sparsity at N=160, but it does *not* make estimates "unstable." Our metrics (e.g., RD) use proportions. A subgroup with **n=0 (proportion p=0%) is a definitive measurement, not 'missing data'**—it is the direct evidence of extreme representational bias.
> > - **Conclusion:** In this context, the sparsity observed in IRIS-Gen-52 is not a failure of our benchmark; it is the flaw in the model that our benchmark has successfully captured. We will, however, seek to increase N from 160 in future iterations to obtain a larger sample size for intersectional subgroups, thereby enhancing the statistical power and credibility of these fine-grained measurements.
> >
> > ## **Q3: Counterfactual Controls**
> >
> > **Response:**
> >
> > - **Gen-Task (BIS-Gen):** The control is absolute. The prompts are text-based, and we only change the single target attribute word (e.g., "...young **light-colo**r skin tone female gymnast." vs. "...young **dark-color** skin tone female gymnast.").
> > - **Und-Task (BIS-Und):** We used **InstructPix2Pix**, which is explicitly designed for **instruction-guided image editing** (modifying a specified attribute in an image while ensuring that the global structure and other features remain unchanged). However, we did not trust it blindly. As stated in §C.2, we *verified* that only the target attribute changed by using (1) the ARES classifier and (2) manual spot-checks, discarding failed pairs. And we will release all generated images to encourage future human-in-the-loop validation.

---

> > > ### Comment · Reviewer_RvCU · 2025-11-25
> > > **Reviewer Response**
> > >
> > > Thank you for your thoughtful response, I truly appreciate the effort you put into addressing my questions and concerns. Most of my concerns have now been resolved. In addition, considering the general author response as well as the responses to the other reviewer, I believe the paper is in good shape for acceptance to ICLR.
> > >
> > > That said, I strongly encourage the authors to use the additional pages wisely to further improve the presentation and readability of the paper (I fully agree with Reviewer Z61h’s concerns regarding clarity and organization of the paper).
> > >
> > > In recognition of the authors’ effort in addressing the raised issues, I have increased my score.

---

> > > > ### Author Response · Authors · 2025-11-27
> > > > **Response to Reviewer RvCU**
> > > >
> > > > # **Response to Reviewer RvCU**
> > > >
> > > > Dear Reviewer RvCU,
> > > >
> > > > We are truly grateful for your decision to raise the score to 8 and for recognizing our efforts during the rebuttal. We sincerely appreciate your constructive suggestion to "use the additional pages wisely to further improve the presentation and readability."
> > > >
> > > > We fully agree with you (and Reviewer Z61h) that the organization of the paper can be improved. Therefore, we have submitted a revised manuscript (with a significant restructuring of  §3) to improve the writing and logical flow:
> > > >
> > > > 1. **Refined Context:** We have condensed the **Introduction** and **Related Work** sections to be more concise, reducing redundancy as suggested.
> > > > 2. **Logical Restructuring:** We have reorganized **§3** (grouping content into: Definitions $\rightarrow$ Implementation $\rightarrow$ Analysis $\rightarrow$ Generalizability Explanation) to prioritize the "Why" before the "How":
> > > >    - **§3.1:** Definitions of the core evaluation dimensions (IFS, RFS, BIS) and metrics.
> > > >    - **§3.2 - §3.4:** Implementation details, including the evaluation scope, pipeline and supporting tools.
> > > >    - **§3.5 - §3.6:** The analytical scoring workflow and qualitative diagnosis (IRIS-MBTI).
> > > >    - **§3.7 (New):** A dedicated discussion on the framework's extensibility.
> > > > 3. **Detailed Extension Guide:** To support the points made in §3.7, we added **§A.4**, providing concrete technical guidelines for future researchers to extend IRIS to new attributes and metrics.
> > > >
> > > > We think this new structure makes the paper much easier to follow. Thank you again for your strong support and valuable guidance!
> > > > If you have any further suggestions, please feel free to let us know.

---

### Official Review · Reviewer_Z61h · 2025-10-27

**Soundness:** 1
**Presentation:** 1
**Contribution:** 2
**Rating:** 4
**Confidence:** 3

**Summary:**

The authors create a multidimensional fairness benchmark based on Ideal Fairness, Real-world Fidelity, and Bias Inertia & Steerability (IRIS) with customized datasets and multiple sensitive attributes. As part of this, they develop Adaptive Routing Expert System (ARES), a classifier for demographics in images. They posit various phenomenon foregrounded by IRIS, including a "generation gap" and "personality splits."

**Strengths:**

1. The authors address an important problem (the diversity of fairness metrics and their practical applicability) and target a viable strategy (a multidimensional benchmark that integrates these metrics in practical contexts).
2. The toolkit appears well-documented and transparent.
3. The authors bring up some interesting concepts, such as personality profiles and the generation gap, that could enrich the fairness literature if developed fully.

**Weaknesses:**

1. The authors are trying to do way too much in a single paper. The core contribution is meant to be a unification of fairness measures, but there is very little on this (e.g., why this set of measures is better than alternatives). The contributions, such as unifying different notions of fairness in a common framework, are not yet developed or defended enough to constitute a major contribution to the fairness literature.
2. The paper is too verbose with many paragraphs and sentences that are just repeatedly stating what the paper contributes instead of actually showing how it advances our understanding. Lines 041-053 is an example, and honestly I think basically the first 4 pages could be greatly shortened (though I like Figure 1 and Table 1). This would free up room to develop scholarly contributions.
3. I don't see why there is an algorithm (Lines 162-129) in this paper. It seems like this is just a way of calculating score across dimensions. An algorithm (and the algorithm box) is not the best way to describe a scoring procedure because there's nothing particularly algorithmic. Presenting it differently would be much clearer, such as by not having undefined variables and unclear notation.
4. The core intellectual content of the paper is in the selection of items in the second and third columns of Table 1, but I'm not convinced of the authors' choices. This is the main part of the paper that I need to see developed before being more positive on the submission.
5. The paper remains focused on particular sensitive attributes (e.g., age) and modalities without clear generalization. Even if my other points are addressed, I'm not sure if this is really a general fairness evaluation strategy rather than just one of potentially many combinations of metrics. This would be fine if the sensitive attributes were just examples of a broader machine learning theory, but that is currently lacking.
6. I find the current ordering of the paper very unclear. I suggest not going by "enabling technologies" but instead chronological through the application of the tool. Start with the datasets and the metrics and how they relate to the fairness literature.

**Questions:**

1. How were the items in the second and third columns of Table 1 determined? For example, why is statistical parity used, when it is widely considered an oversimplified fairness metric? I understand it is mathematically convenient (hence it being used so often), but is it sufficient for real-world application?
2. Why are IFS and RFS separated in that way? What exactly do they mean? I encourage the authors to speak in terms of the fairness literature (e.g., "statistical parity" is a well-defined concept that is widely understood in the literature) rather than trying to coin new terminology. For example, "fairness through awareness" is very broad and plausibly includes everything else in the second column except "fairness through unawareness," which is a largely outdated notion that doesn't actually seem to show up in the empirics of the paper.

Also see the Weaknesses section.

---

> ### Author Response · Authors · 2025-11-19
> **Detailed Response to Reviewer Z61h**
>
> # **Detailed Response to Reviewer Z61h**
>
> We sincerely thank Reviewer Z61h for the sharp and candid critique. Your feedback made it clear we failed to fully *articulate* our core intellectual contribution in the initial draft—a failure of expression, not of contribution. We are grateful for this opportunity to clarify our defense.
>
> ## **W1, W4, Q-all: Insufficient Defense of Core Contribution**
>
> **Response:** This is your most central concern. Please see **Point 1 of our General Response** for a detailed defense of:
>
> 1. **Our Core Contribution:** We offer a "dashboard" (a unified *framework*) for trade-off analysis, not a "single unified metric" (a *unification* of measures).
> 2. **The Ternary Chain Philosophy:** We defend why IFS ("defaults"), RFS ("cognition"), and BIS ("execution") are the necessary and complementary axes for evaluating UMLLMs.
>
> Here, we add specific replies to your sub-points:
>
> - **On Statistical Parity (SP):** We agree SP is "oversimplified" as a *normative* goal. Our framework uses it as a *Diagnostic Baseline*. We *descriptively* measure deviation from a "default factory setting" precisely to contrast it with the philosophically conflicting RFS.
> - **On "Fairness through Awareness" & IFS/RFS separation:** RFS is the *Operationalization* of "Awareness" (cognitive accuracy). We *must* separate IFS (pursuing "should-be" equality) and RFS (pursuing "as-is" accuracy) because they represent the two primary, conflicting philosophical poles. Quantifying this exact trade-off is a core contribution of IRIS.
> - **On BIS:** This is the *Operational Dimension*. If IFS/RFS diagnose the model's static state, BIS assesses its dynamic *plasticity* and answers the critical intervention question: "Is the model teachable to be fair?"
> - **On Extensibility:** Our **General Response (Point 1 (3))** also details how our framework's extensibility (open space, robust normalization, modular ARES) addresses "completeness" concerns.
>
> We trust this more detailed demonstration of our "design philosophy" directly addresses your critique of the "core intellectual content."
>
> ## **W2, W3, W6: Verbose Narrative and Organization**
>
> **Response:** We thank you for this feedback and agree that conciseness is critical. We acknowledge your concerns about the introductory verbosity and the "Algorithm 1" presentation.
>
> Given the rebuttal's time constraints and customs, we have prioritized adding critical experiments (Sensitivity Analysis, ARES performance analysis and Prompt Robustness) in the current PDF revision to ensure technical soundness and version stability.
>
> We have noted this for the camera-ready to preserve the review integrity of the current page limit/structure, and **commit to addressing these presentation concerns in the final Camera-Ready version**, where we will shorten the introduction and remove repetitive statements. We are open to any specific suggestions you might have on which paragraphs could be most effectively condensed.
>
> ## **W5: Generalizability**
>
> **Response:** Your concern is valid. We clarify: IRIS is a generalizable **framework**; our paper presents one **instance**. The attributes we chose (age, gender, skin tone) are among the most common in the literature and are well-supported by datasets, but the IFS/RFS/BIS methodology and the high-dimensional fairness space work flow themselves are **extensible**. Researchers can apply, and we will also attempt in future, it to new attributes (e.g., disability, emotion) or dimensions (e.g., Causal Fairness) by (1) defining subgroups and metrics, (2) adding new classifier experts to the modular ARES system, and (3) providing new real-world data for RFS. This is also discussed in our **General Response (Point 1)** on "Extensible Design."

---

> > ### Comment · Reviewer_Z61h · 2025-11-19
> >
> > I thank the author for their response. I will retain my score, in part because the author response largely repeats points made in the paper and the same jargon or idiosyncratic terminology, rather than substantively changing the paper or persuasively rebutting my concerns. I still think the paper lacks significant contribution to the fairness literature and is therefore marginally below the acceptance threshold.

---

> > > ### Author Response · Authors · 2025-11-27
> > > **Response to Reviewer Z61h**
> > >
> > > # **Response to Reviewer Z61h**
> > >
> > > Thank you for your continued engagement. We genuinely appreciate your critique regarding the paper's clarity and organization.
> > >
> > > We have taken your feedback seriously and have uploaded a revised manuscript with substantial changes to address your concerns directly:
> > >
> > > 1. **Conciseness:** We have significantly condensed the **Introduction** and **Related Work** to remove repetitive statements, focusing the narrative on core contributions.
> > > 2. **Reordered Methodology:** Following your insight that the previous ordering was unclear, we have completely restructured **§3** to follow a more natural narrative arc:
> > >    - **Theory First (§3.1):** We now start with the *design rationale* of the dimensions and metrics.
> > >    - **Tools & Implementation (§3.2-3.4):** We then introduce the evaluation scope, supporting tools (ARES, Datasets) and the execution pipeline.
> > >    - **Analysis (§3.5-3.6):** The scoring algorithm and MBTI diagnosis are now presented as the analytical method *after* the metrics are defined, and additional detailed explanations were added to the algorithm to improve the clarity.
> > > 3. **Concrete Generalizability (New §3.7 & §A.4):** To demonstrate the framework's extensibility beyond the current attributes, we added a new subsection (§3.7) and a detailed **§A.4**, which serves as a technical manual for extending IRIS to new domains.
> > >
> > > We think these concrete structural changes have significantly improved the manuscript's readability and logic. Thank you for reminding us to further refine the presentation.

---

### Official Review · Reviewer_LWHZ · 2025-10-30

**Soundness:** 3
**Presentation:** 4
**Contribution:** 4
**Rating:** 8
**Confidence:** 3

**Summary:**

This paper introduces IRIS, a benchmark for measuring fairness in unified multimodal large language models (UMLLMs) across both understanding and generation. IRIS evaluates their fairness across three dimensions: Ideal Fairness, Real-World Fidelity, and Bias Inertia & Steerability. The framework uses a custom demographic classifier (ARES) and four curated datasets to compute 60 fairness metrics projected into a “fairness space.” It also includes a qualitative profiling tool (IRIS-MBTI) that summarizes a model’s fairness personality.
Through this benchmark, the authors uncover systemic phenomena—like the generation gap (UMLLMs being less fair in generation), personality splits (task-dependent fairness behavior), and the counter-stereotype reward (improved output quality when breaking stereotypes).

**Strengths:**

• Novel and meaningful contribution: A first benchmark to jointly assess fairness in UMLLMs. The unified fairness-space idea and the three-dimensional structure are elegant and address the “Babel Tower” problem of conflicting fairness metrics. The ARES classifier and supporting datasets make multi-dimensional fairness evaluation feasible in an automated fashion.

• Clarity: The paper is easy to follow despite its scope. Figures clearly walk the reader through the pipeline, and the IRIS-MBTI profiles make results intuitive and interpretable. I appreciate the figures!

• Interesting findings. The “generation gap” and “personality split” results are particularly interesting and likely to influence future multimodal fairness research.

**Weaknesses:**

* Conceptual trade-off: Fairness is inherently multi-dimensional, and each metric captures a distinct philosophical or statistical notion. Collapsing these / projecting them into a high-dimensional “fairness space”  might blur important nuances and make interpretability harder. How the authors balance this abstraction with human-understandable fairness judgments would be worth discussing.

* Metric sensitivity. The benchmark combines 60 metrics with tuned weights. It’s unclear how sensitive the global IRIS score is to metric choice or weighting; an ablation would help establish robustness.

**Questions:**

1. How is the IRIS-MBTI profiling validated? Are these categories empirically meaningful or mainly heuristic?

2. How robust are the IRIS scores to metric weighting or selection changes?

3. How do the authors think about the potential loss of nuance or interpretability when collapsing many distinct fairness notions?

---

> ### Author Response · Authors · 2025-11-19
> **Detailed Response to Reviewer LWHZ**
>
> # **Detailed Response to Reviewer LWHZ**
>
> We are sincerely grateful to Reviewer LWHZ for the enthusiastic support, particularly for noting our framework as "elegant" and our presentation as "excellent" (4/4).
>
> ## W1, Q3: On Conceptual Trade-offs and Loss of Nuance
>
> **Response:** We fully agree. The risk of **"blur[ring] important nuances"** by **"collapsing"** metrics is precisely the 'Babel Tower' problem our IRIS framework was designed to solve.
>
> We wish to clarify that the core contribution of IRIS is **not** the single overall IRIS-score, but rather the **six-dimensional score vector** (the "dashboard" of IFS-Und, RFS-Gen, etc.). This vector *preserves* the distinctions between the "philosophical or statistical notions" you mentioned (e.g., IFS vs. RFS) and *forces* a trade-off analysis, rather than optimizing for a single, collapsed solution.
>
> The "loss of nuance" risk you rightly identify is an *implementation* challenge, not a *conceptual* one, which we address directly:
>
> - **Conceptually (Lossless):** Traditional aggregation methods (such as using averages or extreme values instead of the whole) may struggle to inherit the perspective/philosophical meaning of the underlying indicators. But in our "fairness space" (§3.1), each of the 11 core metrics (60 granular metrics) is an independent axis. A model's "position" in this space theoretically retains all information from all metrics. Different regions of this space may imply different fairness behaviors, which is a key area for our future exploration.
> - **Implementation (The Risk):** The danger is that differing metric scales (e.g., [0, 1] vs. [0, $\infty$]) could cause small-scale metrics to be "drowned out" when computing the L2 distance (the final score).
>
> This is **exactly why we designed our robust normalization pipeline** (§ A.2.3). We *deliberately avoided* data-driven min-max or z-score normalization. Instead, we use **normalization by theoretical maximums** (for bounded metrics) and **log-transforms** (for unbounded ones). This design ensures that all metrics—and their "philosophical notions"—make a proportional and meaningful contribution. No perspective is "muted" by its mathematical scale.
>
> In short: The "collapsing" problem is what our **high-dimensional fairness space solves**. The implementation risk of this collapse (varying scales) is what our **normalization strategy mitigates**.
>
> ## W2, Q2: On Metric Sensitivity and Robustness
>
> **Response:** Please view details in **General Response (Point 3)**, which details our new Leave-One-Out (LOO) sensitivity experiments and manuscript revisions.
>
> ## Q1: On IRIS-MBTI Validation
>
> **Response:** We confirm the latter: the IRIS-MBTI is a heuristic diagnostic tool, not an empirically validated psychometric instrument. Its value is its "qualitative summary" capability (§3.2), translating a complex 6D vector (e.g., "High-IFS, Low-RFS, Low-BIS") into an intuitive, anthropomorphized label/snapshot (e.g., UDR - "The Dogmatic Preacher..."). It complements the quantitative scores by providing a rapid, high-level guide. We have clarified its heuristic nature in the main text and Appendix (§3.2, A.3) and guided readers on how to interpret and utilize this tool in Conclusion (§ 6).

---

> > ### Comment · Reviewer_LWHZ · 2025-11-19
> >
> > I acknowledge that I have read the authors' rebuttal. I have no significant concerns with this work.

---

> > > ### Author Response · Authors · 2025-11-27
> > > **Response to Reviewer LWHZ**
> > >
> > > # **Response to Reviewer LWHZ**
> > >
> > > Dear Reviewer,
> > >
> > > We sincerely appreciate your recognition of our efforts and your consistent support.
> > >
> > > Inspired by the discussion phase, we have uploaded a final revised manuscript to further enhance readability.
> > >
> > > We have optimized the narrative flow of **§3** **Methodology** (grouping content into: Definitions $\rightarrow$ Implementation $\rightarrow$ Analysis) and condensed the **§Introduction** and **§Related Work**. Additionally, we added **§A.4** to provide detailed guidelines for extending the framework, further solidifying its contribution to the community.
> > >
> > > We hope these improvements further justify your positive assessment. Thank you!

---

### Official Review · Reviewer_ci6e · 2025-10-31

**Soundness:** 3
**Presentation:** 3
**Contribution:** 3
**Rating:** 8
**Confidence:** 3

**Summary:**

This work tackles the ever-present and frustrating problem in fairness that there are many, sometimes conflicting, fairness metrics that can be used; however, ML practitioners often pick only one metric to judge a model’s fairness.  The authors present a comprehensive fairness benchmark for unified multimodal large language models (UMLLMs) called IRIS.  They simultaneously analyse the fairness of the understanding and generative capabilities of a model by incorporating 11 fairness metrics into a novel embedding space across 3 dimensions to measure what they call Ideal Fairness, Real-world Fidelity, and Bias Inertia and Steerability.  They analyse a number of modern UMLLMs to demonstrate the utility of the benchmark and highlight a number of phenomena such as a gap in fairness across generation and understanding of models.  They conclude by showing how the benchmark can be used to understand where bias is generated within a model to allow for effective mitigation techniques.

**Strengths:**

1. The benchmark is a valuable contribution to the community and demonstrates that you can combine numerous, sometimes conflicting, fairness metrics into a single benchmark for a holistic view of the fairness of UMLLMs.
2. The benchmark analyses both understanding and generative capabilities of the models.
3. The authors demonstrate effectively the utility of the benchmark for determining characteristic of models and phenomena that might be present.
4. The authors demonstrate how you might use the benchmark to understand the source of bias within models and use that the mitigate biases going forward.
5. The contribution of annotated datasets for the ARES classifier.
6. The tests to determine the foundational integrity of the benchmark are a great addition to this paper.

**Weaknesses:**

1. Too much of the paper is in the Appendix.   I don’t know how this can be fixed as the paper has a lot of content, but a huge proportion of the paper is in the Appendix, which was too long for me to evaluate completely.  The authors should consider ways in which you can bring some important details into the main paper concisely (like the models analysed and the personality profiles).  Many important assumptions are also only reported in the appendix (the simplification of gender, age, and skin tone attribute sub-groups for example).  Any important considerations, limitations, or assumptions should be expressed in the main body of the paper, even if they are expanded on in the Appendix.
2. Biases in the ARES classifier will propagate into the metrics which is impossible to disentangle from the biases in the models being measured.
3. I like the idea of personality profiles for models, but it would be good to have names that are more easily consumable and intuitive than UAF, HDF, etc.  explained in the main paper in a table.
4. This was mentioned in the limitations and is fairly significant “The scoring pipeline depends on calibrated hyper-parameters whose robustness across other model families and domains requires further validation”.

**Questions:**

1. Line 069: Figure 1 link to models being tested is incorrect.  Should be A.1.1.
2. How can you guarantee that biases from the ARES classifier don’t propagate into the benchmark measurements?  Have you measured its bias?  How might we reduce this risk going forward?  It has an overall accuracy of 88%.  Is this sufficient?  Where was it failing?  Do you propagate the uncertainty generated by an imperfect classifier into the benchmark results so that consumers know which results are reliable?
3. Can you please state in the main paper what demographic markers the ARES classifier classifies (line 305)?
4. Line 046 and 126: the literature on the bias transfer hypothesis [1] is somewhat divided on the conclusion that intrinsic biases “carry over” to downstream tasks.  Studies in the fine-tuning space are conflicted with some works saying that biases do transfer [1,2,3], while others claim they do not [4].  Recent work fixes some issues with previous studies, uses unified metrics to measure intrinsic and extrinsic biases, and demonstrates that they do conclusively transfer across prompting adaptation [5].  It’s a minor thing but the authors should update the text to reflect the state of the community on this.
5. Line 086: In reference to a fairness assessment goal seeking a single optimal solution - isn’t it the case though that sometimes there is one fairness metric that represents the ideals that the ML practitioners are trying to adhere to?  So this isn’t always a negative and maybe as part of this benchmark you would also want to report the individual fairness metrics as well.
6. How do you obtain 60 “granular” metrics from the 11 metrics in table 4?  Line 159 say they are derived from the core 11 metrics but I don’t see in App A.2.1 how they are expanded to 60 metrics.
7. Line 174: how is m_i normalised to form u_i?
8. Line 230: I don’t understand the Core Question & Metrics for Understanding in Real-world Fidelity.  Can you please rephrase this to make it clear what the metric is for this?  This sounds more like a research question to answer rather than a metric.
9. Line 373: can you comment on the practical implications of the trade-off between Real-world Fidelity and Steerability?
10. Line 375: How are Real-world Fidelity (gen) and Ideal Fairness (understanding) synergistic?  They are correlated which means their combined effect might not be greater than the sum of their parts.
11. Line 383: Could you please expand on how you determined that the two traits HAF and UDF across generation and understanding means that “This finding suggests that a shared representation space does not guarantee consistent fairness characteristics across tasks”?
12. Line 429: There is a typo in the heading “Gudied” -> “Guided”.
13. Line 457: Interesting finding.  But I wonder if the internal representations are just a result of slightly OOD data and this is a form of “over-fitting” phenomena where large but opposing weights are used to model noisy data.
14. What effect does the suite of metrics chosen have on the framework?  Have you done a sensitivity analysis to determine how sensitive the method is the choice of core fairness metrics?
15. In the Ethics statement on line 492 you should also mentioned that the use of the ARES annotator can introduce bias as well as measurement noise.
16. More discussion on why the three core dimensions of Ideal Fairness, Real-world Fidelity, and Bias Inertia and Steerability, were picked would have been appreciated.


[1] Ryan Steed, Swetasudha Panda, Ari Kobren, and Michael Wick. 2022. Upstream Mitigation Is Not All You Need: Testing the Bias Transfer Hypothesis in Pre-Trained Language Models. In Proceedings of the 60th Annual Meeting of the Association for Computational Linguistics, Dublin, Ireland.
[2] Sarah Schröder, Alexander Schulz, Philip Kenneweg, and Barbara Hammer. 2023. So can we use intrinsic bias measures or not? In Proceedings of the 12th International Conference on Pattern Recognition Applications and Methods.
[3] Masahiro Kaneko, Danushka Bollegala, and Naoaki Okazaki. 2022. Debiasing Isn’t Enough! – on the Effectiveness of Debiasing MLMs and Their Social Biases in Downstream Tasks. In Proceedings of the 29th International Conference on Computational Linguistics, Gyeongju, Republic of Korea.
[4] Xisen Jin, Francesco Barbieri, Brendan Kennedy, Aida Mostafazadeh Davani, Leonardo Neves, and Xiang Ren. 2021. On Transferability of Bias Mitigation Effects in Language Model Fine-Tuning. In Proceedings of the 2021 Conference of the North American Chapter of the Association for Computational Linguistics: Human Language Technologies, Association for Computational Linguistics.
[5] Nivedha Sivakumar, Natalie Mackraz, Samira Khorshidi, Krishna Patel, Barry-John Theobald, Luca Zappella, and Nicholas Apostoloff, Bias after Prompting: Persistent Discrimination in Large Language Modelsm, EMNLP, 2025.

---

> ### Author Response · Authors · 2025-11-19
> **Detailed Response to Reviewer ci6e (Part 1 of 2)**
>
> # **Detailed Response to Reviewer ci6e**
>
> We are grateful to Reviewer ci6e for the insightful review, positive assessment and for recognizing our work as a "valuable contribution".
>
> ## W1 & W3: Moving content to main body / MBTI names not intuitive
>
> **Response:** We agree and have moved essential content to the main text:
>
> 1. A brief overview of the evaluated models
>
>    > (Lines 245-248)
>
> 2. Our demographic attribute definitions
>
>    > (Lines 249-258)
>
> 3. The IRIS-MBTI personality archetype table
>
>    > (now Table 2, Lines 270-282)
>
> Furthermore, we clarify that the codes (UAF, HDF) were inspired by the MBTI typology (like INFJ). The new Table 2 now maps these codes to their intuitive names (e.g., "The Adaptive Idealist") and provides clear descriptions, which we agree significantly improves readability.
>
> ## W2, Q2, Q3, Q15: ARES classifier bias and performance
>
> **Response:** Please view details in **General Response (Point 2)**. Our revisions include:
>
> 1. Clarifying classified attributes (Age, Gender and Skin Tone) in the main text
>
>    > (Line 342-347)
>
> 2. Adding a comprehensive Appendix B.4  with fine-grained performance and fairness (AD) analysis
>
>    > (Lines 1566-1711)
>
> 3. Updating the Ethics Statement to mention annotator noise and bias
>
>    > (Line 546)
>
> We trust this provides a much clearer picture of our classifier's reliability.
>
> ## W4: Hyperparameter robustness
>
> **Response:** Please view details in **General Response (Point 3)**. As discussed in Appx. A.2.4 ("Limitations of Manual Calibration"), the S/K hyperparameters only affect readability and do not alter the absolute model rankings. These results demonstrate that our core findings are robust.
>
> ## Q1, Q12: Typos
>
> **Response:** Thank you for catching these. We have corrected all typos.
>
> > (Lines 069, 493)
>
> ## Q4: Bias Transfer literature
>
> **Response:** Thank you for these references. We have updated our Introduction and Related Work to more accurately reflect the complex consensus on the bias transfer hypothesis.
>
> > (Lines 046, 47); (Lines 126-130)
>
> ## Q5: Value of single fairness metrics
>
> **Response:** You are correct that practitioners sometimes care about just one metric. Our framework enriches single-metric analysis, rather than replacing it. It allows a user to inspect a single metric (e.g., [M1] RD) while simultaneously revealing its trade-off cost (e.g., in RFS). We already showed per-metric data in §D and commit to releasing all raw granular values for public analysis.
>
> ## Q6: From 11 to 60 granular metrics
>
> **Response:** Thank you for the clarification request. The 60 metrics are the 11 core metrics (Table 5) applied across various attribute intersections (L1: Age, L2: Age + Gender, L3: Age + Gender + Skin Tone). For example, [M2] Accuracy Disparity is calculated at 7 different subgroup levels. We have clarified this derivation in A.2.1 and provided a full list in the new Table 6.
>
> > (Line 979-1020)
>
> ## Q7: Normalization of m_i to u_i
>
> **Response:** As detailed in §A.2.3: (1) Bounded metrics (e.g., RD) use theoretical maximum normalization ($u=m/m_{max}$), (2) Unbounded metrics (e.g., penalties) use a log-transform ($u=\log(1+P)$) to prevent outlier dominance. We have added a direct pointer from Algorithm 1 to this explanation.
>
> > (Line 174)
>
> ## Q8: RFS-Understanding Core Question
>
> **Response:** Thank you. The original "How to probe..." described our methodological challenge (testing internal knowledge decoupled from visual input). We have revised Table 1 (line 195) to be a clear question about the model (**Does internal knowledge (decoupled from visual input) reflect real-world demographic facts?**), as you suggested.
>
> > (Line 195)
>
> ## Q9: RFS/BIS trade-off implication
>
> **Response:** This strong negative correlation ($\rho = -0.80$) is a key **statistical** finding from our benchmark.  Our analysis suggests that models scoring high on RFS-Gen are adept at reproducing complex statistical priors from their training data, which often stems from real-world collection and thus inherently reflects "As-is" societal biases. By definition, the BIS-Gen task challenges these priors by demanding counter-stereotypical images.
>
> Our data appears to suggest a tension in current architectures between high fidelity to priors and flexibility for counter-factuals. A possible explanation is that high-fidelity models (e.g., $P(\mathrm{female}|\mathrm{nurse}) \gg P(\mathrm{male}|\mathrm{nurse})$) may suffer a generation success rate (GSR) drop when prompted for a low-probability combination ('male nurse').
>
> We agree that the root cause (data, architecture, or training objective) is a critical direction for future research, and the value of IRIS is in revealing this phenomenon through quantitative means.

---

> ### Author Response · Authors · 2025-11-19
> **Detailed Response to Reviewer ci6e (Part 2 of 2)**
>
> ## Q10: RFS(gen) / IFS(und) synergy
>
> **Response:** This positive correlation ($\rho = 0.57$) is an encouraging phenomenon.
>
> RFS-Gen measures the ability to reproduce statistical priors derived from the real world, while IFS-Und measures the ability to ignore irrelevant priors and make a fair judgment based on visual evidence (e.g., correctly identifying a 'nurse' in an image, even if it's a male nurse). These two evaluation dimensions are not inherently conflicting, allowing the possibility for a model to excel in both.
>
> This synergy seems to suggest a positive possibility: models with strong RFS-Gen may, in turn, be better at making accurate judgment in understanding tasks. We believe that investigating models that overlap on these metrics is a valuable direction for future research.
>
> ## Q11: HAF/UDF "personality split"
>
> **Response:** This is a crucial point, and we agree this requires more detailed explanation. Our core point is that this is an **observation** and **hypothesis** based on our experimental results, **not a theoretically** proven conclusion. Our derivation is as follows:
>
> **I. Clear Personality Difference:** Our benchmark (IRIS-MBTI) identified a 'personality split' in the VILA-U model. These two personalities represent opposing fairness tendencies:
>
>    **HAF (The Heuristic Reformer):** In the **Understanding** task, VILA-U scores *low* on Ideal Fairness (IFS) but *high* on Real-world Fidelity (RFS).
>
>    **UDF (The Grounded Reformer):** In the **Generation** task, VILA-U shows the opposite pattern: *high* IFS and *low* RFS.
>
> **II. Phenomenon vs. Assumption:** Crucially, we observe an **opposition** between the model's behavior on the IFS and RFS axes across the two tasks, despite both tasks relying on the same core representations.
>
> **III. Implication for Shared Representation:** The architectural premise of UMLLMs is their unified, cross-task (generation and understanding) shared visual-linguistic representation space. Intuitively, one might expect this shared foundation to lead to *consistent* fairness characteristics.
>
> **IV. Preliminary Conclusion:** The behavioral 'split' described in **Point II** explicitly fails to satisfy the intuitive assumption of consistency outlined in **Point III**. We therefore suggest that a shared representation space does not guarantee consistent fairness characteristics across tasks. This implies that task-specific components (e.g., different decoders, projection layers, or training objectives) may have a massive, even "reversing", impact on the final fairness outcome.
>
> This is a preliminary finding that highlights the necessity of our dual-task evaluation framework, and we believe it is highly valuable for future research, which we will validate with broader statistics and deeper mechanistic probes.
>
> ## Q13: "Counter-stereotype reward" as OOD
>
> **Response:** This is a rational hypothesis. While 'OOD' is a plausible factor, our internal probes (§5.2, E.2) indicate that counter-stereotypical prompts elicit consistently higher 'energy' (L2 Norm) and 'complexity' (Participation Ratio). This suggests that these challenging samples induce the model to engage in more extensive computation with higher-dimensional activations. We will explore this interesting phenomenon further in future work.
>
> ## Q14: Metric selection sensitivity
>
> **Response:** Please view details in **General Response (Point 3)**. Beyond our original weight-floating experiment (Fig. 4(b)), we have added a new, stricter Leave-One-Out (LOO) analysis (new Appx. A.2.2). The framework is highly robust: removing any single metric still results in $\rho > 0.928$.
>
> > (Lines 1026-1109)
>
> ## Q16: Why these three dimensions?
>
> **Response:** Please view details in **General Response (Point 1)**, which provides our detailed defense for the IFS/RFS/BIS framework.

---

> > ### Comment · Reviewer_ci6e · 2025-11-25
> >
> > I'd like to thank the authors for their detailed response.  I feel they have addressed the majority of my concerns and that the work is now in an excellent state and provides a strong contribution to the community.  For this I have raised my score.

---

> ### Author Response · Authors · 2025-11-27
> **Response to Reviewer ci6e**
>
> # **Response to Reviewer ci6e**
>
> Dear Reviewer,
>
> Thank you again for your strong support and positive evaluation of our work, and your recognition of our efforts during the rebuttal.
>
> To ensure the final paper meets the highest standards of clarity and to address the structural feedback raised during the discussion phase, we have uploaded a polished version of the manuscript.
>
> We have optimized the narrative flow of **§3** **Methodology** (grouping content into: Definitions $\rightarrow$ Implementation $\rightarrow$ Analysis) and condensed the **§Introduction** and **§Related Work** (Crucially, while streamlining the text, we have retained and integrated all the key literature and recent advancements you previously highlighted, ensuring the paper accurately reflects the current state of the field). Additionally, we added **§A.4** to provide detailed guidelines for extending the framework, further solidifying its contribution to the community.
>
> Thank you for your time and support!

---

### Author Response · Authors · 2025-11-19
**General Response to All Reviewers (Part 5 of 5)**

## **3. Benchmark Robustness and Sensitivity Analysis**

Reviewers raised important concerns about the benchmark's robustness to the selection of **hyperparameters** (specifically scaling factors $S$ and $K$) and **granular metrics** ($m_i$).

### **1) Hyperparameter Sensitivity**

**(Responding to Reviewer LWHZ and Reviewer ci6e)**

Our benchmark's rankings are highly robust because our core methodology deliberately avoids data-driven hyperparameters.

- **Normalization:** We explicitly avoided model-sensitive min-max or z-score normalization. Instead, we use normalization by **theoretical maximums** or **log-transforms** (see Appx. A.2.2). This ensures our benchmark is "open" and "additive," allowing new models and metrics to be added without changing the baseline.
- **Scaling (S and K):** The *only* hyperparameters used (S and K) are for the final step: scaling the Euclidean distance (the objective fairness measure) for **readability** into an approximately 0-100 range, which is easily understood by humans. These parameters **do not, under any circumstances, alter the relative ranking of models.**

While rankings are robust, we acknowledge (A.3.4) that the "perceptible range" of scores may shift as models improve, and we commit to updating these scaling guidelines to maintain readability.

### **2) Metric Sensitivity**

**(Responding to Reviewer LWHZ and Reviewer ci6e)**

Our original paper (Fig. 4(b)) already conducted the Metric Sensitivity experiment by: 1) adjusted the weights of the sub-metric sets for the three dimensions (IFS, RFS, and BIS) by 10% when calculating the final total score; and 2) adjusted the weights of select sub-metrics within each of the three dimensions by 10%. Parts of the results, as shown in Figure 4(b), demonstrated that after these adjustments, the Spearman’s $\rho$ of the new model rankings against the original was > 0.96.

To address this concern more thoroughly, we conducted a new, stricter, fine-grained **Leave-One-Out (LOO) sensitivity analysis**. We iteratively removed *each* of the 60 granular sub-metrics, recalculated the total IRIS score, and re-ranked the models.

The results were definitive: the Spearman rank correlation remained exceptionally high, with **$\rho > 0.9286$ in all 60 cases**. Even the lowest correlation (0.9286, from removing Penalty_$\Delta$GSR) confirms the final rank is not dependent on any single metric. The full LOO analysis is now in the appendix.

These experiments powerfully demonstrate that our aggregation method is robust and the final rankings represent a holistic consensus, not a dependency on any single metric.

### **3) Manuscript Revisions**

- **Added New Appendix Section (A.2.2):** We added a new section detailing this rigorous Leave-One-Out metric sensitivity analysis and its results.

  > (Lines 1026-1109)

- **Updated Main Text (§4.1):** We updated the text in §4.1 to reference this new, more comprehensive sensitivity analysis.

  > (Lines 356, 357)

---

### Author Response · Authors · 2025-11-19
**General Response to All Reviewers (Part 4 of 5)**

## **2. Quantifying ARES Classifier Error and Uncertainty**

We are grateful for the critical concerns regarding the ARES classifier. We agree that a biased or inaccurate ruler would invalidate the benchmark, and this component demands the strictest scrutiny.

In response, we conducted a fine-grained performance decomposition of ARES and an analysis of its internal biases.

### **1) On "Is 88% accuracy sufficient?": Fine-grained Performance Decomposition**

**(Responding to Reviewer ci6e)**

The reviewer's concern about the 88% overall accuracy is reasonable.

First, we emphasize that 88% is, in itself, a strong result. Classifying *generated* occupational portraits poses unique challenges not found in standard real-world datasets (e.g., FairFace, UTK-Face), as the classifier must handle numerous interferences (see **new Fig. 9**), such as:

- Common generative artifacts (e.g., distorted, occluded, or incomplete facial features).
- Artistic processing (e.g., unique art styles, strong confounding lighting).

As detailed in Appendix C.4, we explicitly trained ARES to handle these interferences. The 88% accuracy was measured on our most challenging *Random-200* dataset, which is replete with such artifacts. Given these challenges, we believe this accuracy demonstrates the robustness of our system compared to standard classifiers which often fail on generated content.

Furthermore, to provide a definitive breakdown, we conducted new experiments on a **new, larger, human-annotated, and balanced V-720 dataset** (N=720, with 40 samples for each of the 18 intersectional attribute combinations).

On this new validation set, ARES demonstrates high accuracy,  consistent with our existing ablation studies :

- **Gender:** 98.9% accuracy (98.5% on Random-200, 99.7% on Selected-300)
- **Age:** 93.2% accuracy (93.0% on Random-200, 97.7% on Selected-300)
- **Skin Tone:** 90.4% accuracy (91.5% on Random-200, 93.1% on Selected-300)

(Full P/R/F1-scores and confusion matrices are in **new Appendix B.4**).

### **2) On "ARES's own bias": Classifier Fairness Analysis**

**(Responding to Reviewer ci6e and Reviewer RvCU)**

A good 'ruler' must also be fair. To investigate if ARES injects bias, we used the balanced V-720 dataset to calculate the classifier's own **Accuracy Disparity (AD)** (Max Recall - Min Recall). This measures whether predictive power is biased toward any group.

The results show ARES's internal AD is extremely low:

- **Gender (0.56%):** Negligible, confirming high fairness.
- **Age (4.17%) & Skin Tone (5.83%):** Very low.

Our analysis reveals this minor disparity is driven by the **"middle" categories** (middle-age, middle-skintone). This is not systemic bias, but an expected statistical property:

- "Extreme" categories (e.g., *young*, *dark*) have *unique* features, forming separable clusters.
- "Middle" categories are an *ambiguous spectrum* with *feature overlap* (e.g., some middle-aged samples look young, some look old), placing them *near decision boundaries* and making them inherently harder to classify.

In summary, this is an expected, non-systemic error that is uniformly distributed across all models, and does not affect the relative ranking between models.

### **3) Manuscript Revisions**

Based on this, we have made the following revisions:

- **Added New Appendix Section (B.4):** We added a new appendix section dedicated to this deep validation of ARES. It contains the full, fine-grained P/R/F1 reports, confusion matrices, the classifier's own fairness (AD) analysis, and a discussion of these limitations with case studies.

  > (Lines 1566-1711)

- **Clarified Main Text (§4.1):** We clarified in §4.1 that the 88% accuracy is the "Overall" score on "challenging data" and added a pointer to the comprehensive analysis in the new Appendix B.4.

  > (Lines 345, 346)

---

### Author Response · Authors · 2025-11-19
**General Response to All Reviewers (Part 3 of 5)**

### **3) On Framework Limitations and Extensible Design**

**(Responding to Reviewer LWHZ and Reviewer Z61h)**

Finally, we address **concerns (R-LWHZ, R-Z61h)** about framework completeness. We agree that AI fairness is a vast, evolving field, and our "default-cognition-execution" chain does not capture all perspectives.

However, IRIS was designed from the ground up with **extensibility** as a core principle to meet this challenge. We ensure this through:

- **Open High-Dimensional Space:** Our core methodology (§ 3.1, Alg 1) projects metrics into an open space. New, computable metrics can be seamlessly incorporated as new dimensions without invalidating the framework.
- **Robust Normalization Strategy:** We deliberately avoid data-driven *min-max* or *z-score* normalization. We use normalization by theoretical maximums or log transforms (§ A.2.2), ensuring score stability and **additivity** for new metrics.
- **Modular ARES Toolchain:** ARES is a modular adaptive expert system. We currently use age, gender, and skin tone as illustrative, widely-used attributes, consistent with broad practice, but new attributes (e.g., "emotion") can be added by simply training a new expert and adding it to the ARES routing network.
- **Swappable Data:** For RFS, we use BLS and Eurostat data as examples. Researchers can substitute this with any real-world distribution matching their target region or context.

In summary, the current limitations of IRIS were the very motivation for its open and extensible design in the future.

### **4) Manuscript Revisions**

Based on this discussion, we have made the following revisions:

- Expanded **§3.2** to detail the "Should-be (IFS) $\rightarrow$ As-is (RFS) $\rightarrow$ Can-be (BIS)" evaluation chain  (i.e., "default $\rightarrow$ cognition $\rightarrow$ execution").

  > (Lines 157-161, 211-213).

- Revised **Table 1** to clarify the "Core Question" for RFS-Understanding, making the objective clearer in response to **R-ci6e**.

  > (Lines 196-199).

- Moved essential content (evaluated models, attribute setup, MBTI archetypes) from the appendix into the main text (**§3.3**) to improve readability.

  > (Lines 244-264, 270-282).

---

### Author Response · Authors · 2025-11-19
**General Response to All Reviewers (Part 2 of 5)**

### **2) The Design Philosophy of IRIS: A Three-Dimensional Evaluation Chain**

**(Responding to Reviewer ci6e and Reviewer Z61h)**

Reviewers critically questioned why we chose these three core dimensions (IFS, RFS, BIS). We defend this choice explicitly in the revised manuscript (**see Table 1 and §3.2**) and summarize the rationale here.

Building on our trade-off philosophy, we selected dimensions that effectively construct this tension-filled fairness space. We argue that fairness evaluation for UMLLMs should be a **complete logical chain**: from its "default instincts" to its "real-world cognition," and finally to its "controllability." The three distinct, non-overlapping axes of IRIS cover this entire chain:

#### Ideal Fairness (IFS): Measuring the Model's "Should-be"

- **Core Question:** In the absence of specific context, what are the model's intrinsic, unconditional "Default Values"? How far do its priors deviate from a Utopian, perfectly egalitarian ("should-be") world?
- **Metric Choice:** This is why we select metrics like Representation Disparity (RD) and Statistical Parity Difference (SPD). **(To R-Z61h):** While SPD may be "oversimplified" as a *normative* goal, we argue it is the *exact* tool for this *diagnostic* purpose: to precisely measure deviation from an "ideal uniform distribution".
- **Practical Implication:** IFS indicates the model's "factory safety settings". High IFS implies a lower risk of "out-of-the-box" harm. **Examples:** 1. Public APIs (where prompts are uncontrolled). 2. Creative/Educational Contexts (e.g., children's books) that desire an idealized world.

#### Real-world Fidelity (RFS): Measuring the Model's "As-is"

- **Core Question:** After assessing the "ideal," we must assess its "reality". Does the model accurately comprehend the world "as-is"?
- **Metric Choice:** This dimension evaluates if model outputs faithfully reflect observable, real-world distributions. It embodies "descriptive awareness" (echoing "Fairness through Awareness" and "descriptive fairness"), assessing the model's knowledge foundation: *does the model know the real-world ratios?*
- **Practical Implication:** RFS indicates "cognitive accuracy". **Examples:** 1. Societal Simulation: Generating synthetic data that accurately reflect demographic distributions for social science research. 2. Decision Support: Applications requiring priors grounded in statistical reality rather than idealized equality.

#### Bias Inertia & Steerability (BIS): Measuring the Model's "Can-be"

- **Core Question:** Given the model's "defaults" (IFS) and "cognition" (RFS), how much *Inertia* does it exhibit? How steerable is it toward a desired, counter-stereotypical ("can-be") state?
- **Metric Choice:** This final "action" step draws from Counterfactual Fairness. We measure *if* the model can be steered (e.g., $\Delta$GSR) and the *cost* of doing so (e.g., performance *Penalty* via QPS, Hallucination Rate).
- **Practical Implication:** This is the model's "teachability" or "alignment cost." A high BIS (low inertia) model may be highly valuable even if its IFS/RFS is poor. **Examples:** 1. AI Alignment: Assessing the difficulty of debiasing a base model during fine-tuning (high inertia = hard to align). 2. High-Fidelity Control: Applications requiring the generation of specific, counter-stereotypical samples (e.g., "an older dark-skin male nurse") while maintaining high image quality.

---

### Author Response · Authors · 2025-11-19
**General Response to All Reviewers (Part 1 of 5)**

# **General Response to All Reviewers**

We are encouraged that **Reviewer ci6e** and **Reviewer LWHZ** rated our work highly (8), recognizing the IRIS framework as a "valuable contribution" and an "elegant" design. We are equally grateful for the constructive feedback from **Reviewer Z61h** and **Reviewer RvCU**. While they raised valid concerns, we appreciate that they also highlighted the importance of our problem setting, the novelty of our unified pipeline, and the value of our empirical findings (e.g., the "generation gap"). We have addressed the core concerns from all reviewers with new experiments and revisions below.



## **1. Core Contribution and Philosophy (IFS/RFS/BIS) :**

### **1) Clarifying the Core Contribution: A "Dashboard," not a "Single Metric"**

**(Responding to Reviewer Z61h)**

We must first clarify our response to **R-Z61h**'s critique that the "core contribution... [is not yet] developed or defended enough".

Our core contribution is **not** to propose a new, single "unification of fairness measures". This is often mathematically infeasible (e.g., the inability to simultaneously satisfy demographic parity, equalized odds, and predictive rate parity, except in trivial cases).

Instead, our core contribution is to provide the first **unified fairness benchmark framework** for various architectures of UMLLMs. The value of IRIS lies precisely in this:

- **Philosophical Innovation: A "Dashboard," not a "Single Metric."** Our work does not attempt to "unify" (i.e., collapse) the multitude of philosophically and mathematically conflicting fairness metrics. Instead, IRIS serves as a **"dashboard"**. Beyond the single overall IRIS-score, we also provide the **six-dimensional score vector** (IFS-Und, RFS-Gen, etc.), as well as the Personality diagnostic. Its value lies in simultaneously displaying these conflicting values (e.g., IFS vs. RFS) to enable **Trade-off Analysis**. Our solution to the "Babel Tower" dilemma is not to find the lost "universal language" but to provide a high-quality "simultaneous interpretation system" to manage these conflicts.
- **Methodological Instantiation: IFS, RFS, and BIS as the Axes of Trade-off.** Our proposed dimensions—Ideal Fairness, Real-world Fidelity, and Bias Inertia & Steerability—are the concrete instantiation of this trade-off analysis.
- **Disciplinary Necessity: Filling the UMLLM Evaluation Gap.** IRIS fills a critical gap in the emerging UMLLM paradigm. Unlike existing unified benchmarks that mainly focus on capabilities (e.g., reasoning), IRIS targets the equally critical domain of **fairness**. Crucially, applying this multi-dimensional analysis has allowed us to uncover systemic issues in leading UMLLMs—such as the **"Generation Gap"** and **"Personality Splits"**—which single-metric evaluations may fail to capture.

---

### Meta-Review · Area_Chair_AdPg · 2026-01-11

**Summary:**

The submission underwent fair amount of discussion between the authors and the reviewers. After the discussion the paper receives overall positive ratings except one reviewer (Z61h, rating 4). But the AC agrees with Z61h's point that (1) sentences are verbose (the AC personally conjecture that the authors use LLMs quite heavily), (2) rationale behind selection of items in the 2nd, 3rd column of Table 1 seems not well discussed and (3) the algorithm box may not be necessary. Despite the flaws, many reviewers appreciate that the contribution of combining many metrics into a single benchmark for UMLLMs. The AC agrees with the majority of the reviewers' appreciation thus recommends to accept the submission in ICLR 2026. For the camera ready version, the AC suggests that (1) removing decorative words (likely by LLMs) (eg, "sound foundation", "rigorously validate" "reliable technical cornerstone") and (2) seemingly larger space between word characters (check your LaTeX template setting).

**Reviewer Concerns:**

Most of the reviewers' concerns are well addressed by authors' terrific responses (except the responses use big fonts to make the AC need to scroll a lot). But the concerns by Z61h are not well addressed and the AC thinks that some of the reviewer's concerns are valid (see meta review). Please address them in the final copy.

**Reviewer Scores:**

All reviewers quite actively participate in the discussion to make the decision process easier.

---

### Decision · Program_Chairs · 2026-01-26

Accept (Poster)